# Deciphering the phase transition-induced ultrahigh piezoresponse in (K,Na)NbO₃-based piezoceramics

Mao-Hua Zhang [1,2,12], Chen Shen[2,12], Changhao Zhao [2,12], Mian Dai[2], Fang-Zhou Yao [3], Bo Wu [4✉], Jian Ma[4], Hu Nan[5], Dawei Wang [5], Qibin Yuan[6], Lucas Lemos da Silva [7], Lovro Fulanović [2], Alexander Schökel [8], Peitao Liu[9], Hongbin Zhang [2], Jing-Feng Li [1], Nan Zhang [10✉], Ke Wang [1,11✉], Jürgen Rödel [2] & Manuel Hinterstein[7]

Here, we introduce phase change mechanisms in lead-free piezoceramics as a strategy to utilize attendant volume change for harvesting large electrostrain. In the newly developed (K,Na)NbO₃ solid-solution at the polymorphic phase boundary we combine atomic mapping of the local polar vector with in situ synchrotron X-ray diffraction and density functional theory to uncover the phase change and interpret its underlying nature. We demonstrate that an electric field-induced phase transition between orthorhombic and tetragonal phases triggers a dramatic volume change and contributes to a huge effective piezoelectric coefficient of 1250 pm V$^{-1}$ along specific crystallographic directions. The existence of the phase transition is validated by a significant volume change evidenced by the simultaneous recording of macroscopic longitudinal and transverse strain. The principle of using phase transition to promote electrostrain provides broader design flexibility in the development of high-performance piezoelectric materials and opens the door for the discovery of high-performance future functional oxides.

[1] State Key Laboratory of New Ceramics and Fine Processing, School of Materials Science and Engineering, Tsinghua University, Beijing, China. [2] Department of Materials and Earth Sciences, Technical University of Darmstadt, Darmstadt, Germany. [3] Center of Advanced Ceramic Materials and Devices, Yangtze Delta Region Institute of Tsinghua University, Jiaxing, China. [4] Physics Department, Southwest Minzu University, Chengdu, China. [5] School of Microelectronics, Faculty of Electronic and Information Engineering, Xi'an Jiaotong University, Xi'an, China. [6] School of Electronic Information and Artificial Intelligence, Shaanxi University of Science and Technology, Xi'an, China. [7] Institute for Applied Materials, Karlsruhe Institute of Technology, Karlsruhe, Germany. [8] Deutsches Elektronen-Synchrotron DESY, Hamburg, Germany. [9] Shenyang National Laboratory for Materials Science, Institute of Metal Research, Chinese Academy of Sciences, Shenyang, China. [10] Electronic Materials Research Laboratory, Key Laboratory of the Ministry of Education and International Center for Dielectric Research, Xi'an Jiaotong University, Xi'an, China. [11] Wuzhen Laboratory, Jiaxing, China. [12] These authors contributed equally: Mao-Hua Zhang, Chen Shen, Changhao Zhao. ✉email: wubo7788@126.com; nzhang1@xjtu.edu.cn; wang-ke@tsinghua.edu.cn

Ferroic materials with coexisting states of comparable energy typically exhibit extraordinary responses to external stimuli, such as giant electrocaloric effect[1,2], ultrahigh energy storage density[3–5], colossal magnetoresistance[6], and giant magnetostriction[7]. This feature can be exploited in poled ferroelectrics to design advanced piezoelectric materials that afford direct electromechanical coupling between mechanical and electrical energy under electric field or mechanical stress. Substantial theoretical and experimental evidence has indeed verified that the highest piezoelectric response is often observed in the phase transition regions[8–10], where active structural/microstructural changes are particularly sensitive to external stimuli[11–15]. While the exact origin of the high piezoresponse of the renowned lead zirconate titanate, $Pb(Zr,Ti)O_3$ (PZT) at the morphotropic phase boundary (MPB) remains a topic of debate, almost all theories point to the critical role of the existence of a phase boundary with coexisting states of comparable energy[16–19]. In bismuth-based piezoelectric materials, giant electromechanical response is attributed to the ergodic relaxor to ferroelectric phase transition in $Na_{1/2}Bi_{1/2}TiO_3$-based system[20–22] and the field-induced phase transition between different symmetries in $BiFeO_3$ system[23].

Piezoceramics represented by PZT have a dominating market position in technological applications such as microelectromechanical systems (MEMS), new-generation wireless sensor networks, medical diagnostics through ultrasound imaging, and actuators in the automotive industry. Unfortunately, the use of lead-containing substances in electronic products causes concerns about lead toxicity and contamination[24], which has led to widespread scientific activities in the search for environmentally-benign alternatives in the last two decades[25–28]. It should be noted that the transfer of lead-free piezoceramics into applications is still in its infancy state[29], partly due to the high versatility of the Pb-based materials[30]. Therefore, it is of utmost importance to gain a deeper insight into the structure-property relationship of lead-free piezoelectrics for the design of high-performance lead-free materials.

Potassium sodium niobate (KNN)-based materials are characterized by a good combination of high piezoelectricity and high operating temperature[31,32] and account for the largest share (nearly 40%) of the lead-free piezoceramics market, which is forecast to grow at an annual rate of >20%[33]. Interestingly, KNN-based materials also exhibit enhanced electromechanical properties at phase boundaries where the coexistence of phases is found. Accordingly, efforts to tailor the electromechanical properties of KNN materials have been focused on phase boundary engineering[34], which aims to construct orthorhombic-tetragonal type polymorphic phase boundaries (PPB)[25,35] and rhombohedral–tetragonal type MPB[36–38]. It is therefore surprising that the fundamental nature of the contribution of the phase boundary to the electromechanical properties, especially for high-performance compositions with a piezoelectric coefficient $d_{33}$ exceeding 350 pC N$^{-1}$ has not yet been systematically investigated[39,40].

To this end, a series of KNN compositions located at the orthorhombic-tetragonal phase boundary with a high piezoelectricity $d_{33}$ up to 510 pC N$^{-1}$ were developed. Our work combines local high-resolution transmission electron microscopy (TEM) with global angle-dependent in situ high-energy synchrotron X-ray diffraction studies on three pre-selected key compositions at and off the phase boundary. Piezoelectric contributions by lattice extension, domain wall movement, and phase transitions can thus be elucidated and are contrasted to a macroscopic volume change obtained from a combination of axial and transverse macroscopic strain. As result, an electric field-induced phase transition between the orthorhombic and tetragonal phases occurs in the optimal composition located at the

boundary but is not significant in the other two compositions that are off the phase boundary. First-principles calculations, supplemented by quasi-isothermal calorimetric measurement were performed to clarify the origin of the phase transition induced by an electric field.

## Results

**Macroscopic properties.** The nominal compositions $(0.97-x)$ $K_{0.50}Na_{0.50}Nb_{0.965}Sb_{0.035}O_3$-$0.03(Bi_{0.5}Na_{0.5})_{0.9}(Ga_{0.5}Li_{0.5})_{0.1}ZrO_3$-$xBiFeO_3$ ($x = 0$, 0.002, 0.004, 0.005, 0.006, 0.007 and 0.008) of the investigated ceramic materials were obtained by the admixture of different amounts of the multiferroic material $BiFeO_3$. The admixture of $BiFeO_3$ constructs the PPB at room temperature by increasing the rhombohedral-orthorhombic phase transition temperature $T_{R-O}$ and decreasing the orthorhombic-tetragonal phase transition temperature $T_{O-T}$ (Supplementary Fig. 1). All compositions exhibit a piezoelectric coefficient $d_{33}$ higher than 330 pC N$^{-1}$, as depicted in Fig. 1a. The highest piezoelectricity of 510 pC N$^{-1}$ is obtained for composition $x = 0.006$, with a $BiFeO_3$ content of 0.6 mol%. Both the nearest compositions, $x = 0.005$ and $x = 0.007$, also feature a high $d_{33}$ of 465 and 420 pC N$^{-1}$, respectively. The piezoelectric coefficient $d_{33}$ of 510 pC N$^{-1}$ with a Curie temperature of 256 °C renders the composition $x = 0.006$ a top-class candidate among all KNN-based piezoceramics.

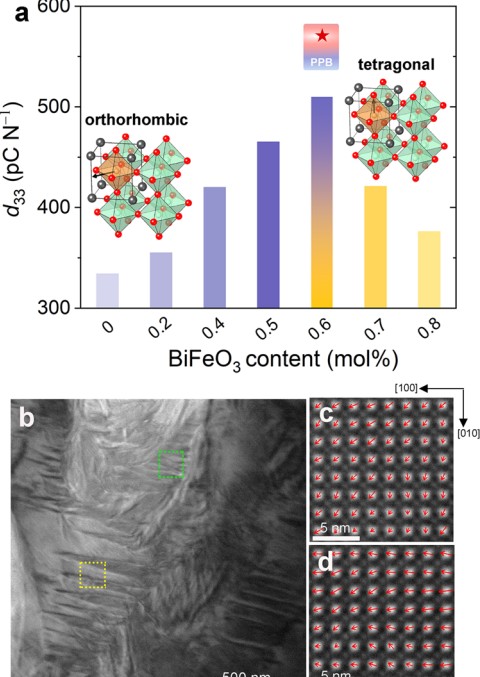

**Fig. 1 Piezoelectricity and characterization of the local structure of the studied KNN materials. a** Piezoelectric coefficient $d_{33}$ of the studied KNN compositions $x = 0$, 0.002, 0.004, 0.005, 0.006, 0.007, and 0.008 (with a $BiFeO_3$ content of 0, 0.2, 0.4, 0.5, 0.6, 0.7, and 0.8 mol%, respectively). The optimal piezoelectricity is obtained at the orthorhombic-tetragonal phase boundary, known as the PPB. **b** High-magnification angular bright-field (ABF) image of composition $x = 0.006$ taken along the [100] zone axis. **c**, **d** Atomically resolved high-angle annular dark-field scanning transmission electron microscopy (HAADF-STEM) image taken along the crystallographic [001] zone axis, obtained from the regions highlighted by green and yellow dashed squares, respectively, in **b**. Mapping of the polar vectors is represented by the red arrows. The positions of the atomic columns of the perovskite A-sites and B-sites can be identified by fitting with two-dimensional Gaussian peaks so that the atomic displacement from the B-site cation to the center of the nearest A-site cations can be calculated.

Improved electromechanical properties of KNN piezoelectric materials are typically achieved at the PPB bridging the orthorhombic and tetragonal phases[32,34,41]. Rietveld refinement of high-energy XRD patterns of composition $x = 0.006$ revealed coexistence of tetragonal and orthorhombic phases in the virgin state (Supplementary Fig. 2).

**Characterization of local structure**. Phase coexistence is also verified at a local scale using TEM. The domain structures of composition $x = 0.006$ consist of both regular stripe domains with a width of ~100 nm (yellow dashed square) and irregular complex domains (green dashed square), as demonstrated in Fig. 1b. High-resolution images of the atomic structures within the two domain configurations were taken from the green and yellow dashed squares and are highlighted in Fig. 1c, d, respectively. The atomic displacements from the B-site cations to the center of their nearest A-site cations were calculated, representing the polar vectors of the unit cell. The direction and length of the arrows represent the orientation and magnitude of the polar vectors, respectively. Most of the polar vectors in Fig. 1c are oriented along the [110] direction and some of the polar vectors in the lower right corner are oriented along the [010] direction. In contrast, most of the polar vectors in Fig. 1d are oriented along the [100] direction. Since the spontaneous polarizations of the orthorhombic and tetragonal phases are along the [110] and [100] directions, respectively, the local symmetry can be evaluated. The above results underpin the coexistence of orthorhombic and tetragonal phases, which is not only observed at the level of long-range ordered average structure but is also confirmed by the atomic-scale structure and polar properties, indicating consistency between structures at different length scales. Damjanovic[10] pointed out that the concept of free energy instability, as well as the presence of structural instability, is a general approach that rationalizes the enhanced electromechanical properties of ferroelectrics. In the following sections, the focus is on understanding the structural origin of the high piezoelectricity of the investigated KNN materials.

**Analysis of the electric field-dependent global structure**. Three selected compositions ($x = 0.002$, $x = 0.006$, and $x = 0.008$) were chosen for detailed structural investigation using in situ high-energy X-ray diffraction. Rietveld refinements of the three compositions in the virgin state revealed that the compositions $x = 0.006$ (Supplementary Fig. 2) and $x = 0.002$ (Supplementary Fig. 3) are characterized by a phase coexistence of tetragonal and orthorhombic phases, while in the composition $x = 0.008$ the tetragonal phase dominates (Supplementary Fig. 4). The refined cell parameters of the three compositions are listed in Supplementary Table 1. Contour plots of the 200 reflections of the three compositions as a function of the electric field are revealed in Fig. 2. The evolution of the 200 reflections with the electric field for the composition $x = 0.008$ is characterized by the intensity interchange of the individual reflections (Fig. 2i, j), which is a typical feature of non-180° domain switching[42,43]. However, the compositions $x = 0.002$ and $x = 0.006$ feature a different behavior, i.e., a discontinuous change in the contour plot is observed near the coercive field, as highlighted in Fig. 2a, b, e, f. The change is particularly significant for composition $x = 0.006$, as highlighted in the evolution of peak positions of the 200 reflections (Fig. 2g). In principle, the shift of the reflection positions can be caused by a field-induced lattice strain or an electric field-induced phase transition. The field-induced lattice strain results from the intrinsic piezoelectric effect and therefore, the corresponding shift of the reflection positions should change continuously with the field. Hence, the dramatic change in the $2\theta$ value near the coercive field cannot

be merely justified by the contribution of the lattice strain alone. Given the coexistence of orthorhombic and tetragonal phases in the composition $x = 0.006$, the dramatic structural instability is likely the result of an electric field-induced phase transition.

The calculated field-dependent evolution of the phase fractions of the orthorhombic and tetragonal phases of the three compositions is featured in Fig. 2d, h, l, which was calculated based on a comprehensive Rietveld analysis with texture and lattice strain models[44]. The compositions $x = 0.002$ and $x = 0.006$ are characterized by a well-defined electric field-induced phase transition between orthorhombic and tetragonal phases. The change in phase fractions is particularly significant for the composition $x = 0.006$, i.e., the phase fraction of the tetragonal phase increases to about 70% at the coercive field and then decreases with an increasing electric field, reaching about 47% at the maximum field (Fig. 2h). Although non-180° domain switching dominates in the composition $x = 0.008$, our model suggests that a slight change in the phase fractions cannot be completely ruled out. In the following sections, we focus on the influence of the electric field-induced phase transition on the functional properties of the three compositions.

As measured for respective phase fractions, the sum of the integrated intensities of the 002 and 200 reflections of the tetragonal ($200_T/002_T$) and orthorhombic ($002_O/200_O$) phases in the 0°-sector and the 45°-sector of composition $x = 0.006$ are depicted in Fig. 3a, b, respectively. In the field range of $0-0.40 \, \text{kV} \, \text{mm}^{-1}$, an increase in the phase fraction of the tetragonal phase and a simultaneous decrease in the phase fraction of the orthorhombic phase are observed in both sectors, signifying a phase transition from the orthorhombic to the tetragonal phase (O→T). Between 0.40 and $0.50 \, \text{kV} \, \text{mm}^{-1}$, a phase transition from tetragonal to orthorhombic (T→O) is observed. Above $0.50 \, \text{kV} \, \text{mm}^{-1}$, no significant change is observed in the 0°-sector, while the phase fraction of the orthorhombic phase continues to increase before reaching a plateau at $1.0 \, \text{kV} \, \text{mm}^{-1}$ in the 45°-sector.

The applied field **E** is almost parallel to **Q** in the 0°-sector, and thus the phase fractions in Fig. 3a are exclusively contributed by grains with <200> parallel to the applied electric field. In the 45°-sector, the 200 reflections are contributed by grains with <200> nearly 45° to the applied field, which can involve the grains whose <110> are parallel to the applied field. Since the <110> direction is the spontaneous polarization direction of the orthorhombic phase, the orthorhombic phase is preferred in the 45°-sector at high fields, as revealed in Fig. 3b.

**Analysis of field-induced strain response**. In order to evaluate the electric field-induced strain response in the three selected KNN compositions, the evolution of the center of mass of selected reflections 110, 111, 200, 201, and 211 with the applied electric field in the 0°-sector was extracted to quantify the macroscopic-like volume-averaged contribution to the piezoelectric coefficients[45,46]. The center of mass of each reflection family is calculated by:

$$M_{hkl} = \frac{1}{I_0} \int I(2\theta_i)(2\theta_i)d(2\theta_i) \tag{1}$$

Here $I(2\theta_i)$ is the diffraction intensity at $2\theta_i$. $I_0 = \int I(2\theta_i)d(2\theta_i)$ is the total integrated intensity around each $hkl$ reflection family concerned. Using the Bragg equation, the $d$-spacings for each reflection family can be calculated as:

$$D_{hkl} = \frac{\lambda}{2\sin\left(\frac{M_{hkl}}{2}\right)} \tag{2}$$

Here $\lambda$ is the radiation wavelength. The electric field-induced strain in [$hkl$]-oriented grains can be calculated from the relative

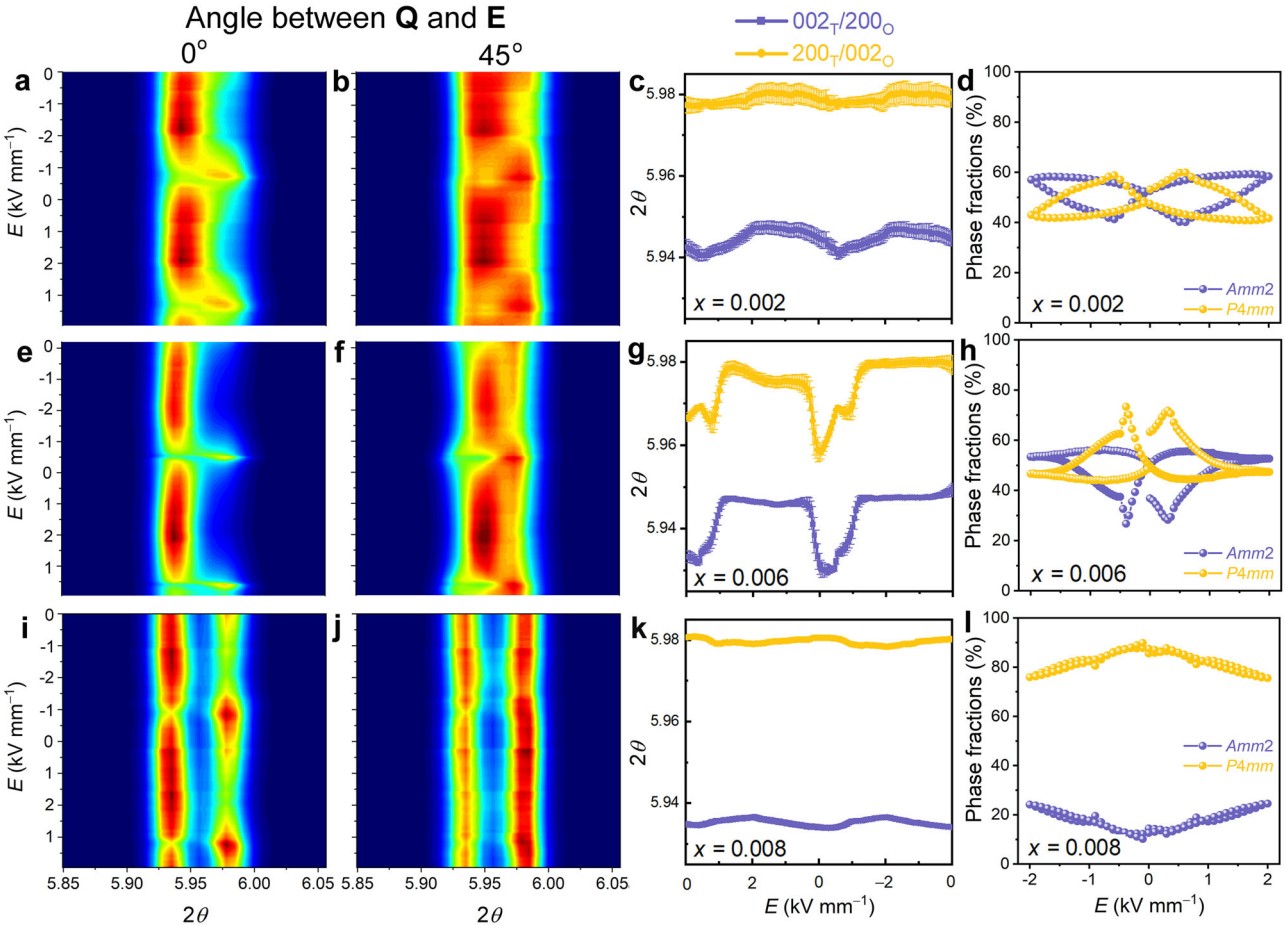

**Fig. 2 In situ contour plots of the 200 reflections with the electric field.** Evolution of the contour plots of the 200 reflections in the 0°-sector and 45°-sector, with a bipolar electric field (0→ 2 kV mm⁻¹ → 0 → −2 kV mm⁻¹ → 0) of **a**, **b** composition $x = 0.002$, **e**, **f** composition $x = 0.006$ and **i**, **j** composition $x = 0.008$. Evolution of the peak positions of the 200 reflections for **c** composition $x = 0.002$, **g** composition $x = 0.006$, and **k** composition $x = 0.008$ with bipolar electric field, obtained from the 45°-sector. Phase fractions of the orthorhombic and tetragonal phases as a function of the applied bipolar field of **d** composition $x = 0.002$, **h** composition $x = 0.006$, and **l** composition $x = 0.008$, calculated using a method that allows the analysis of all strain mechanisms with a Strain, Texture, and Rietveld Analysis for Piezoceramics (STRAP)[44]. The wavelength of the used X-rays is 0.20714 Å.

shift in the mass center $D_{hkl}$:

$$S_{hkl} = \frac{D_{hkl\,E} - D_{hkl\,E=0}}{D_{hkl\,E=0}} \qquad (3)$$

where $D_{hkl\,E}$ is the d-spacing as a function of the applied bipolar field, $D_{hkl\,E=0}$ is the d-spacing at zero field. The strain values calculated from the 110 and 200 reflections of the compositions $x = 0.002, 0.006$, and $0.008$ are displayed in Fig. 3d, e, respectively. Note that the positive strain (at maximum field) of the composition $x = 0.006$ (with 0.100%) is larger than that of composition $x = 0.002$ (0.092%) and composition $x = 0.008$ (0.087%). In particular, the negative strain of the composition $x = 0.006$ (with −0.108%) is significantly larger than that of the compositions $x = 0.002$ (−0.082%) and $x = 0.008$ (−0.062%). The calculated strain loops using the 200 and 110 reflections agree well with the macroscopic properties, as demonstrated in Fig. 3f.

The calculated field-induced strain $S_{hkl}$ from Eq. (3) allows the evaluation of the effective coefficients of the piezoelectric tensor $d_{hkl} = \partial S_{hkl}/\partial E$ (in units of pm V⁻¹) for all grain orientations. Results for compositions $x = 0.002, 0.006$, and $0.008$ are highlighted in a polar diagram in Fig. 3c, where the angular coordinates represent the angle between the applied field and the <200> pole of the grains, and the radial coordinates represent the directional piezoelectric coefficient. The piezoelectric coefficients

$d_{hkl}$ of the composition $x = 0.006$ are significantly larger than those of compositions $x = 0.002$ and $x = 0.008$ in all orientations considered. In particular, the maximum piezoelectric coefficient of composition $x = 0.006$ is achieved in the <200> directions (~1250 pm V⁻¹) is about twice the respective values for the compositions $x = 0.002$ and $x = 0.008$ (~600 pm V⁻¹ in the <200> directions). The huge $d_{200}$ of the composition $x = 0.006$ is due to the field-induced phase transition between orthorhombic and tetragonal phases, occurring over a narrow excursion of the field parallel to <200> (Fig. 2h). Interestingly, $d_{111}$ is only slightly larger for the composition $x = 0.006$ than that for the compositions $x = 0.002$ and $x = 0.008$. The low $d_{111}$ is due to the fact that the <111> direction deviates strongly from both the polar axis of the orthorhombic phase <110> (35.26°) and the tetragonal phase <100> (54.74°) and thus there is limited structural instability (neither O→T nor T→O phase transition).

**Interpretation of the phase transition by density functional theory (DFT) calculations.** To gain further insight into the nature of the phase transition between the orthorhombic and tetragonal phases, first-principles calculations were performed, explicitly taking into account the external electric fields. To facilitate the calculations of the enthalpies and the volume

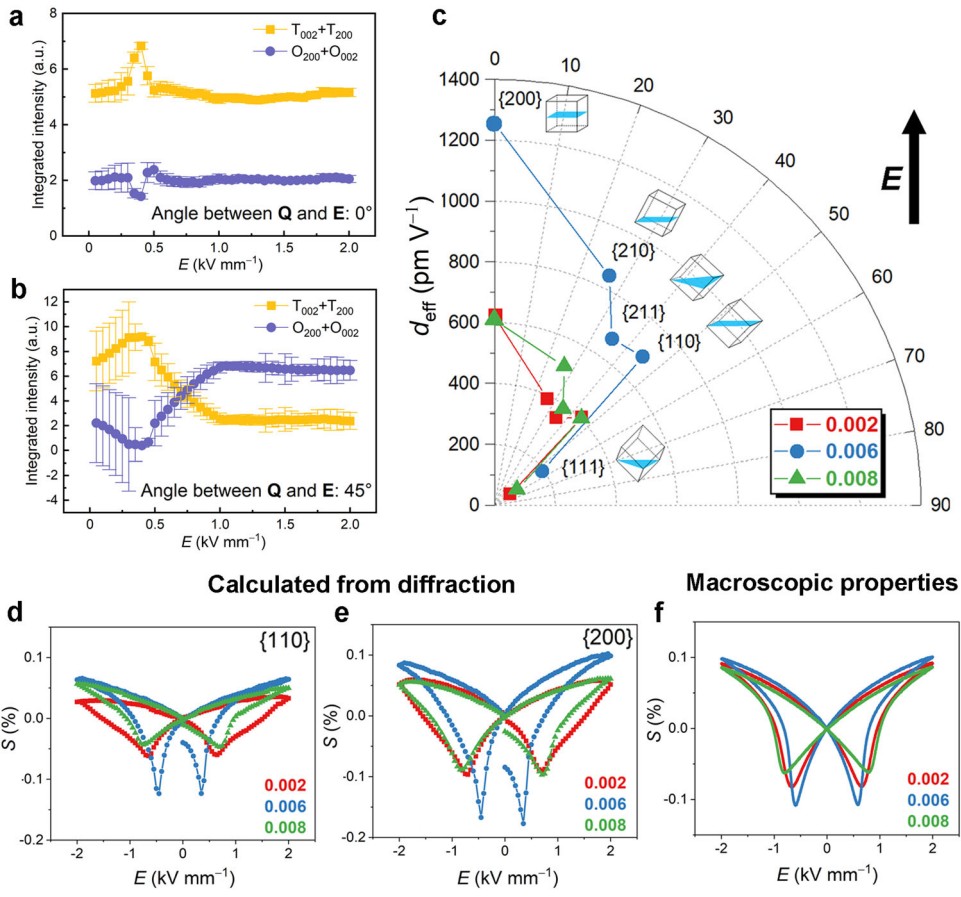

**Fig. 3 Crystallographic analysis of the phase transition and electric field-induced strains. a**, **b** Summed integrated intensity of the 002 and 200 reflections of the tetragonal phase and the orthorhombic phase. Results were obtained in the **a** 0°-sector and **b** 45°-sector at an applied electric field of 0 to 2 kV mm$^{-1}$. **c** Orientation-dependent piezoelectric coefficient $d_{eff}$ of compositions $x = 0.002$, 0.006, and 0.008. Strain-electric field (*S-E*) hysteresis loops calculated from the evolution of the centers of mass of **d** 110 and **e** 200 reflections with the electric field. **f** Experimentally measured macroscopic *S–E* hysteresis loops.

changes of the two phases as a function of the electric field, we consider a particular scenario using rotated supercells such that the spontaneous polarization of the orthorhombic and tetragonal phases are aligned, as featured in Fig. 4a. Electric fields are applied either parallel or antiparallel to the direction of the spontaneous polarization, as illustrated by the arrows in Fig. 4b. When electric fields are applied, the polarization changes in both the tetragonal and orthorhombic phases due to the resulting ionic displacements, resulting in an additional enthalpy contribution. Thus, to quantify the relative stability of the two phases, the enthalpy at 0 K is formulated as follows:

$$F = \varepsilon_0 - VP \cdot E \quad (4)$$

where $\varepsilon_0$ is the ground state energy, $V$ is the unit cell volume, $P$ is the macroscopic polarization, and $E$ is the magnitude of the electric fields. The ground states of the two phases under finite electric fields were calculated self-consistently in the projector augmented wave formalism[47–49] with results featured in Fig. 4b. The unit cell volume of the orthorhombic phase is found to be slightly larger at zero field, the calculated ground-state energy of the orthorhombic phase is 10 meV f.u.$^{-1}$ lower than that of the tetragonal phase, which is even smaller than the thermal fluctuation at room temperature (~26 meV). This slight difference could explain the experimentally observed coexistence of the tetragonal and orthorhombic phases. When an external electric field is applied parallel to the direction of spontaneous polarization, the unit cell volumes of both phases increase with increasing

electric fields, while the unit cell volumes shrink when the applied field is antiparallel to the direction of spontaneous polarization. The calculated enthalpies in the current setting (Fig. 4a) ascertain that the orthorhombic phase becomes more stable when the electric fields are applied parallel to the direction of spontaneous polarization (left half of Fig. 4b), while the tetragonal phase becomes more stable when a reverse electric field is applied (right half of Fig. 4b). Moreover, the enthalpy of the tetragonal phase becomes smaller than that of the orthorhombic phase when the antiparallel electric field becomes stronger than a critical field of 0.006 V Å$^{-1}$, as highlighted by the red dot in Fig. 4b. That is, a structural phase transition between the tetragonal and orthorhombic phases can be induced by the applied electric fields. Such a phase transition corresponds to a volume shrinkage of 0.22% in the DFT calculation (68.56→68.41 Å$^3$ f.u.$^{-1}$), as indicated by the red arrow in Fig. 4b. Therefore, we suspect that such electric field-driven phase transitions can also take place in polycrystalline KNN materials.

The volume change as predicted by the DFT calculation was contrasted to the macroscopic volumetric strain ($S_V$) of the three compositions by simultaneously recording the longitudinal strain $S_{33}$ and transverse strain $S_{11}$ induced by the electric field, as provided in Fig. 4c. Volume shrinkage of 0.05% is observed for the composition $x = 0.006$ near the coercive field, where the electric-field-induced phase transition is particularly significant. The experimentally observed volume change is smaller than the prediction of the DFT calculation,

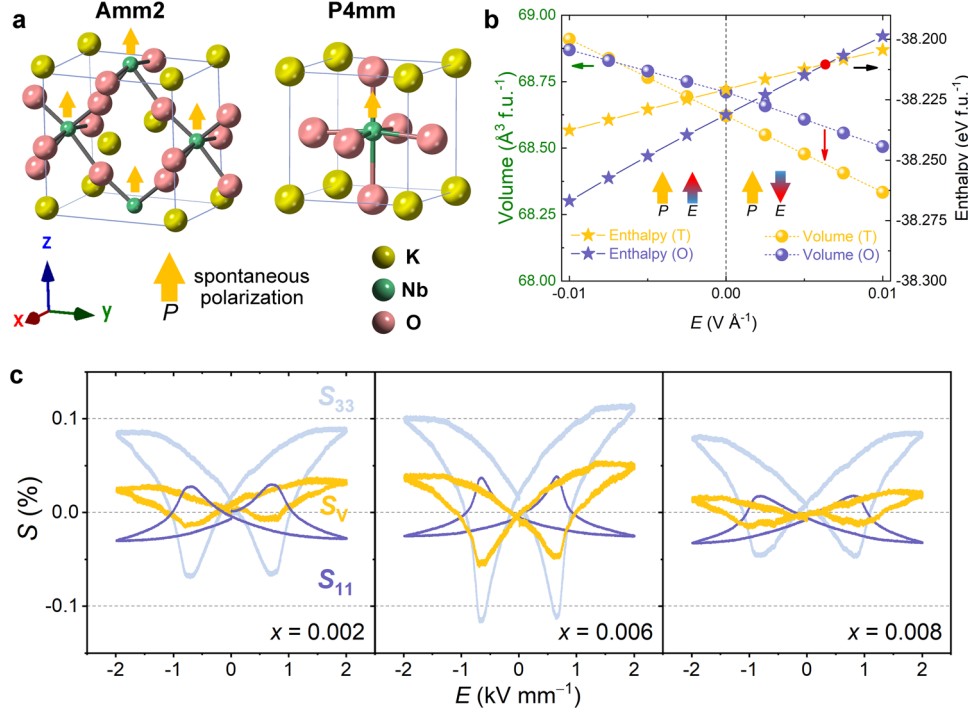

**Fig. 4 DFT calculations and experimental volume changes. a** Schematic of the applied finite electric fields in the DFT calculations. **b** Field dependence of the primitive cell volumes and enthalpies of the tetragonal and orthorhombic phases of $KNbO_3$. The red dot marks the point where the enthalpies of the tetragonal and orthorhombic phases are equal. The red arrow indicates the volume shrinkage for a phase transition from the O phase to the T phase. **c** Experimentally measured macroscopic volume change as a function of the electric field. Simultaneously recorded longitudinal ($S_{33}$) and transverse ($S_{11}$) strains and calculated volume strain of the compositions $x = 0.002$, $x = 0.006$, and $x = 0.008$.

which is expected due to the strain incompatibility between neighboring grains[50].

The volume change in Fig. 4c can be directly contrasted with the unipolar strain. If we assume that the volume change due to phase transition provides a higher strain in direction of the applied electric field, then Fig. 4c asserts that this volume change plays a pivotal role in the large obtained piezoelectric coefficient, as featured in Fig. 1a, note also that the maximum volume change in the composition $x = 0.006$ (at the phase boundary) is about a factor of 2–3 larger than in the other two compositions quantified.

To gain a deeper insight into the phase transition between the orthorhombic and tetragonal phases, we used the climbing image nudged elastic band[51,52] method to find the transition path between the two phases without considering electric fields. The phase transition from the orthorhombic phase to the tetragonal phase is characterized by a smooth and continuous change in the orientation of the spontaneous polarization accompanied by a gradual change in the O-Nb-O bond angle and the O-Nb bond length, as highlighted in Fig. 5. The bond angle and bond length are 166.29° and 2.033 Å in the orthorhombic phase. As Nb gradually shifts during the transition, the bond angle increases, while the bond length decreases. The bond angle and bond length are 180° and 1.861 Å in the tetragonal phase. We note that there is no energy barrier to overcome for such a phase transition, as revealed by the continuous energy curve in Fig. 5. It is known that a first-order transition is characterized by an energy barrier between two end phases and discontinuous order parameter change. Therefore, it is assumed that the phase transition between the orthorhombic and tetragonal phases observed in this work may be a second-order phase transition, without considering the application of an electric field. Moreover, the evolution of the phase fractions changes continuously with the field, as featured in

Fig. 2d, h. The continuous behavior agrees well with the calculated phase transition path characteristic of a barrier-free phase transition between the two phases. The exception in the coercive field is due to the presence of strong structural instability, where dramatic changes in both the structure and functional properties are observed.

To clarify the nature of the phase transition, the temperature change of composition $x = 0.006$ is determined when an electric field is applied since a first-order phase transition and a second-order phase transition can be distinguished by the presence of latent heat and the associated change in sample temperature. An electric field of $1.5\,kV\,mm^{-1}$ was slowly applied to the ceramic sample to avoid the temperature change caused by the electrocaloric effect[53–55], which was poled in the opposite direction, and the electric field was held constant at the maximum electric field to allow time for the system to relax back to ambient temperature, as depicted in Fig. 5b. The first anomaly with a slight negative temperature change is related to the phase transition from the low-symmetry orthorhombic phase to the high-symmetry tetragonal phase where the latent heat is absorbed[56]. It maximizes at $0.37\,kV\,mm^{-1}$, which matches perfectly with the coercive field where the phase transition is the most significant (Fig. 2h). The second anomaly with positive temperature change is related to a combination of domain switching with heat generation due to friction of moving domain walls[57] and the transition from the tetragonal phase to the orthorhombic phase (Fig. 3b). Note that a well-defined field-induced first-order phase transition is characterized by a very sharp change in sample temperature[58,59]. However, the observed temperature change is suppressed and smeared out, which is typically observed in the vicinity of the critical point, where a crossover from first-order to second-order phase transition takes places[60–62]. It has been demonstrated that polycrystalline

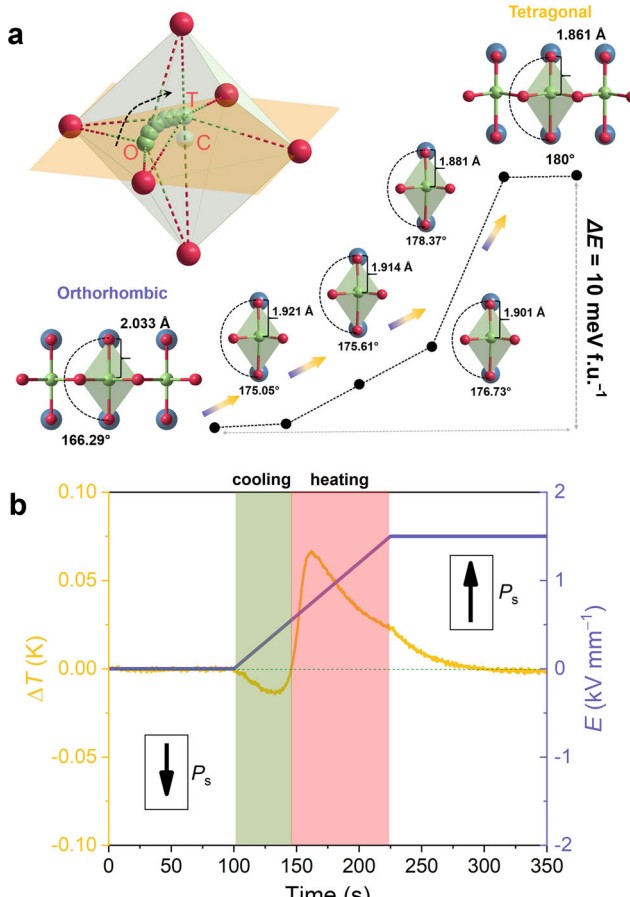

**Fig. 5 Analysis of the phase transition. a** Schematic representation of the phase transition between the orthorhombic and the tetragonal phases of $KNbO_3$ and the corresponding energy levels. **b** Quasi-isothermal calorimetric measurements at the field-induced phase transition of composition $x = 0.006$ at room temperature. A thermistor with a resolution within 0.1 mK was attached to the tested ceramic sample to detect the temperature change. The purple line describes the time dependence of the linearly changing bias electric field and the yellow line provides the change of sample temperature.

materials feature a diffuse critical point due to their inherent inhomogeneity in grain orientation, and are therefore not expected to have a sharply-defined critical point as in single crystals[58]. Therefore, although the absorption of latent heat is observed, the second-order nature of the field-induced phase transition cannot be completely ruled out.

Structural quantification of a range of newly developed high-performance lead-free piezoceramics has ascertained that a phase transformation on the local as well as the global scale is key for high-performance piezoelectricity. The attending phase transformation has been demonstrated to be accompanied by a significant volume change through the macroscopic determination of longitudinal and transverse strain. As a result, the macroscopic volume change is up to 5 times larger than that in the composition without phase transition and the highest effective piezoelectric coefficient of 1250 pm V$^{-1}$ is reached in the <200> directions. Experimental work was supported by first-principles calculations demonstrating that the energy profiles of the two phases are very close and that the tetragonal phase is favored because of the lower enthalpy due to the application of a reverse electric field to the spontaneous polarization. Our study has demonstrated why the design of phase coexistence is of importance to obtain high-

performance lead-free KNN piezoceramics and, more importantly, how structural instabilities at the phase boundary can be harvested to achieve an overall improved electromechanical response of the studied materials. Considering that not all boundary compositions guarantee the acquisition of optimal functional properties (e.g., composition $x = 0.002$ characterized by orthorhombic-tetragonal coexistence), our study also suggests that elaborate composition engineering is required for the identification of the critical composition in polycrystalline materials. This strategy of harnessing phase changes provides a path to a new paradigm for developing high-performance piezoceramics.

## Methods

**Sample preparation**. Piezoceramics with nominal composition $(0.97-x)$ $K_{0.50}Na_{0.50}Nb_{0.965}Sb_{0.035}O_3-0.03(Bi_{0.5}Na_{0.5})_{0.9}(Ga_{0.5}Li_{0.5})_{0.1}ZrO_3-xBiFeO_3$ ($x = 0$, 0.002, 0.004, 0.005, 0.006, 0.007 and 0.008) were synthesized using the solid-state reaction route. Raw chemicals $Na_2CO_3$ (99.8%, Sinopharm, China), $K_2CO_3$ (99%, Sinopharm, China), $Li_2CO_3$ (98%, Sinopharm, China), $Nb_2O_5$ (99.5%, Sinopharm, China), $Sb_2O_3$ (99%, Sinopharm, China), $ZrO_2$ (99%, Sinopharm, China), $Bi_2O_3$ (99%, Sinopharm, China), $Ga_2O_3$ (99.99%, Sinopharm, China) and $Fe_2O_3$ (99%, Sinopharm, China) were weighed stoichiometrically and ball-milled for 24 h in a plastic jar using ethanol as the dispersion medium. Dried mixtures were calcined at 850 °C for 6 h and were milled again for 12 h. The calcined powder with a binder of 8 wt% polyvinyl alcohol (PVA) was pressed into green pellets with a diameter of 10 mm under 10 MPa uniaxial pressure. The pellets were sintered under 1085 °C for 3 h in the air after burning off the PVA at 600 °C for 12 h. For all electrical characterizations, the pellets were ground to a thickness of 0.5–1.0 mm, followed by 400 °C stress-free annealing and sputtering with platinum electrodes.

**Electrical characterization**. Ceramic pellets were polarized in a silicone oil bath at room temperature under a DC field of 4 kV mm$^{-1}$ for 20 min. Piezoelectric coefficient $d_{33}$ was subsequently quantified using a quasi-static piezoelectric constant meter (ZJ-3A, Institute of Acoustics, China). Macroscopic polarization hysteresis loops were obtained using a modified Sawyer-Tower circuit. An electric field of 2 kV mm$^{-1}$ was applied with a frequency of 0.1 Hz. Simultaneously, an optical displacement sensor (D63, Philtec Inc., USA) was used to record macroscopic strain hysteresis loops. Both electric field-induced longitudinal strain $S_{33}$ and transverse strain $S_{11}$ were obtained using linear variable differential transformers[63] and afforded computation of macroscopic volume change ($S_V = S_{33} + 2S_{11}$).

**Latent heat**. A thermistor with a resolution within 0.1 mK was attached to the tested ceramic sample to detect the temperature change. The electric field was slowly increased at a rate of 0.012 kV mm$^{-1}$ s$^{-1}$) in order to get rid of most of the electrocaloric heat generated by the continuous polarization changes to the surroundings, thus allowing observations of the existing latent heat.

**Transmission electron microscopy**. TEM and scanning transmission electron microscopy (STEM) specimens were prepared by mechanically polishing to approximately 20 μm, followed by argon-ion milling in a Gatan PIPS691. TEM and STEM studies were conducted using FEI Titan Cubed Thermis G2300, equipped with double spherical aberration (Cs) correctors. The atomic-scale high-angle annular dark-field (HAADF) images were recorded with a probe size in 9 mode, convergence semi-angle of 25.6 mrad, and collection semi-angle of 67–200 mrad. The A-site and B-site cation positions were determined simultaneously with picometer precision using the two-dimensional Gaussian fitting method. The relative displacement between a B-site cation and its four nearest neighboring A-site cations was calculated and the polarization vectors were extracted. The whole process was automatically completed using APPSA software.

**Synchrotron XRD**. In situ synchrotron X-ray diffraction (XRD) experiments were performed at the P02.1 beamline at the Deutsches Elektronen-Synchrotron (DESY) in Hamburg, Germany. This beamline provides high-energy X-rays with a photon energy of around 60 keV with a radiation wavelength of 0.20714 Å. XRD data were collected in transmission geometry on a specially tailored setup, using a 16-inch two-dimensional (2D) flat panel detector (XRD 1621 NES Series, PerkinElmer, USA)[64]. The detector has 2048 × 2048 pixels and a pixel size of 200 × 200 μm². For in situ electric field-dependent measurements, a bipolar electric field of 2 kV mm$^{-1}$ was applied to the sample, with the electric field direction perpendicular to the incident beam direction. For an entire bipolar electric field cycle, 161 diffraction images, with an exposure time of 10 s for each pattern, were acquired using stepwise changing field values, resulting in an effective frequency of 0.625 mHz.

**Synchrotron data analysis**. The two-dimensional XRD patterns were divided into 18 azimuthal sectors ("cakes") of 5° angle widths and converted into one-dimensional patterns by integrating the intensity in each cake using software Fit2D[65] to obtain intensity against $2\theta$ values. The scattering vector **Q** of each sector

has a different angle to the applied field vector **E**. These sectors are denoted by $X°$-sector ($X = 0$–90) hereafter, where $X$ represents the approximate angle between the corresponding **Q** and **E**.

The 200 reflections (hereafter the $hkl$ indexes are based on the pseudocubic coordinate system with the unit vector $|\mathbf{a}| \approx 4$ Å) of the composition $x = 0.006$ were fitted with a four-peak model with pseudo-Voigt profile using LIPRAS[66] to estimate the change in volume fraction of the orthorhombic phase and tetragonal phase as a function of field amplitude. The four-peak profile consists of the 200 and 002 reflections from the orthorhombic phase ($200_O$ and $002_O$) and the tetragonal phase ($200_T$ and $002_T$). The peak fitting was performed at the 0°-sector and the 45°-sector, as depicted in Supplementary Fig. 7. The initial positions of these 200 reflections were determined from a Rietveld refinement of the virgin pattern of the composition $x = 0.006$ (Supplementary Fig. 2), using a two-phase model consisting of tetragonal and orthorhombic phases. To reduce the number of free fitting parameters, the positions of these individual 200 reflections were fixed in each azimuthal sector, assuming that the peak shift caused by field-induced lattice strain is negligible compared to that caused by field-induced phase transition. Moreover, the mixing parameters of the pseudo-Voigt function were also constrained to be the same for each individual peak. The phase fractions of the tetragonal phase and orthorhombic phase were estimated by the sum of the integrated intensities of corresponding individual reflections.

To evaluate the magnitude of the electric field-induced strain in differently-oriented grains, the evolution of the center of mass of selected reflections 110, 111, 200, 201, and 211 with the applied electric field in the 0°-sector was extracted. The strain as a function of the electric field would carry information on the volume-averaged contribution to the piezoelectric properties. Note this is the only data analysis option that allows quantification of the response of the material to the applied electric field if it is collectively contributed by lattice strain, domain switching, and phase transition but the limited experimental resolution makes it difficult to differentiate between different structural/microstructural mechanisms[45]. The coefficients of the piezoelectric tensor, $d_{hkl}$ represents the calculated piezoelectric coefficient in the $hkl$-oriented grains, and can be calculated by performing linear fit of strain-electric field ($S_{hkl}$–$E$) hysteresis loops in a small range in the vicinity of zero electric field ($d_{hkl} = S_{hkl}/E$).

**STRAP analysis**. The Rietveld, texture, and strain analysis of the full diffraction patterns was performed with the STRAP method[44], using the MAUD program package (Materials Analysis Using Diffraction)[67]. The full orientation series of the caked 2D images with different electric field values were refined with a two-phase structure model of a tetragonal $P4mm$ phase and an orthorhombic $Amm2$ phase. Atomic positions and Debye–Waller factors were refined with the data of the unpoled sample and later kept fixed during the electric field-dependent refinements. A linear background interpolation function between non-overlapping points was refined. The STRAP method consists of a WSODF (Weighted Strain Orientation Distribution Function) model for the field-induced strain and an exponential harmonics model for the field-induced texture. Details can be found elsewhere[44,68,69]. During the electric field-dependent refinement scale factors, lattice parameters, texture, and strain parameters were refined. Phase fractions were calculated based on the scale factors of the coexisting phases.

**DFT calculation**. DFT computations were performed using the projector augmented wave (PAW) method as implemented in the VASP code[70,71], where finite electric fields were considered explicitly by minimizing the electric enthalpy functional[49]. The pseudopotentials with the valence electron configurations of $3s^23p^64s^1$ (K), $4p^65s^14d^4$ (Nb), and $2s^22p^4$ (O) were used. The Perdew–Burke–Ernzerhof (PBE)[72] approximation was employed for the exchange-correlation functional, with a plane-wave energy cutoff being 600 eV. The crystal structures were relaxed with forces on each atom less than 0.005 eV Å$^{-1}$ using a $k$-point density of 0.15 Å$^{-1}$, together with the lattice constants optimized iteratively for all the considered magnitudes of electric fields. Due to limited computational power, we focus on the ferroelectric $P4mm$ (tetragonal, T phase) and $Amm2$ (orthorhombic, O phase) of KNbO$_3$, as similar behavior is expected for (K,Na) NbO$_3$ based on the phase diagram[31]. Initial calculations were performed to validate this assumption against experimental values. To this end, based on the modern Berry phase theory of polarization[73–75], the spontaneous polarization of both phases were determined up to an integer multiple of the polarization quantum $e\mathbf{R}\,\Omega^{-1}$, where $e$ is the electron charge, $\mathbf{R}$ is a lattice vector in the direction of polarization and $\Omega$ is the volume of the unit cell. The calculated macroscopic polarization along the $[001]_{PC}$ direction is $(37.9 + n197.3)$ (where $n$ is an integer) $\mu$C cm$^{-2}$ for the tetragonal phase (Supplementary Fig. 8a) and $(40.8 + n233.2)\,\mu$C cm$^{-2}$ for the orthorhombic phase (Supplementary Fig. 8b). The calculated value is comparable to a macroscopic polarization of 40 $\mu$C cm$^{-2}$ reported in K$_{0.5}$Na$_{0.5}$NbO$_3$ single crystals[76]. In the ground state with a zero electric field, the calculated cell parameters of the tetragonal and orthorhombic phases are $a = 4.027$ Å, $c = 4.217$ Å and $a = 4.016$ Å, $b = 5.821$ Å, $c = 5.858$ Å, respectively, which are in good agreement with the refined cell parameters of KNN (Supplementary Table 1) and results reported elsewhere[77].

**Reporting summary**. Further information on research design is available in the Nature Research Reporting Summary linked to this article.

## Data availability

More relevant data sets generated during and/or analyzed during the current study are available from the first authors and corresponding authors on reasonable request. Source data are provided with this paper.

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

## Acknowledgements
This work was supported by the Basic Science Centre Program of NSFC (No. 51788104), National Nature Science Foundation of China (No. 52032005), National Key Research and Development Program of China (No. 2020YFA0711700), and Beijing Natural Science Foundation (No. JQ20009). N.Z. acknowledges the support of the National Natural Science Foundation of China (No. 12161141012). B.W. acknowledges the support of the National Natural Science Foundation of China (No. 51702028), and the Fundamental Research Funds for the Central Universities, Southwest Minzu University (No. 2020NTD03). M.H. thanks the *Deutsche Forschungsgemeinschaft* (DFG) for funding under Grant No. HI 1867/1-2. M.-H.Z., M.D., and H.Z. are supported by the LOEWE project FLAME funded by the Ministry of Higher Education, Research and the Arts (HMWK) of the Hessen state (Germany). M.-H.Z. expresses his sincere thanks to Professor J. Rödel and Professor K. Wang for their kind and helpful support in continuing this work during his doctoral studies in Darmstadt. We acknowledge DESY (Hamburg, Germany), a member of the Helmholtz Association HGF, for the provision of experimental facilities. Parts of this research were carried out at PETRA III using beamline P02.1. Beamtime was allocated for proposal I-20191018.

## Author contributions
M.-H.Z. and K.W. conceived the idea of this work. B.W. prepared the (K,Na)NbO₃-based ceramic materials with the support of J.M. C.S. and M.D. performed the DFT calculations under the supervision of H.Z. and P.L. M.H. and M.-H.Z. performed the in situ synchrotron XRD measurements with the assistance of L.L.d.S. and A.S. M.-H.Z., C.Z., and M.H. analyzed the in situ synchrotron data and discussed in detail with N.Z. F.-Z.Y. and Q.Y. are responsible for the TEM investigation and received helpful assistance from H.N. and D.W. for the processing of the TEM data. M.-H.Z. and B.W. conducted all electrical

measurements and L.F. finished the latent heat measurement. M.-H.Z. analyzed the structure-property relationship with the help of C.Z. M.-H.Z. drafted the first version of the manuscript and all authors participated in the writing of the paper. K.W. and M.Z. guided the projects. K.W., J.-F.L., and J.R. provided financial and technical support for the accomplishment of this work and provided helpful suggestions.

## Competing interests

The authors declare no competing interests.
