## [Peer Review File · Nature Communications]

Deciphering the phase transition induced ultrahigh piezoresponse in (K,Na)NbO₃-based piezoceramicsREVIEWER COMMENTS

Reviewer #1 (Remarks to the Author):

The paper reports results of a synchrotron X-ray diffraction experiment and a density functional theory calculation of an electric field-induced phase transition of $(\text{K,Na})\text{NbO}_3$ -based lead-free piezoceramics with piezoelectric coefficients higher than 330 pC/N. The authors found that the phase transition between orthorhombic and tetragonal phases triggers a dramatic volume change and contributes to the high piezoelectricity at the polymorphic phase boundary. The results are informative and important to improve and develop high-performance lead-free piezoelectric materials. However, electric field-induced phase transitions at phase boundaries are well-known and have been extensively studied in other piezoelectric materials. For example, field-induced phase transitions in $0.70\text{Pb}(\text{Mg}_{1/3}\text{Nb}_{2/3})\text{O}_3-0.30\text{PbTiO}_3$ piezoceramics have been demonstrated by in situ high-energy x-ray scattering (Phys. Rev. B 97, 214102 (2018)). In addition, the solid solution system reported in this study has a very complex chemical composition based on the known $(\text{K,Na})\text{NbO}_3$ piezoelectric materials, and hence lack the novelty. I recommend that the paper should be published in another specialized journal such as Phys. Rev. B.

Reviewer #2 (Remarks to the Author):

This is a good report which addressed the field-induced extrinsic piezoelectric responses in perovskite-type ferroelectrics. It used state-of-the-art characterization methods to clearly reveal the origin of the high piezoelectric response and the importance of the co-existence of phases of different symmetries with a composition near a phase boundaries. I recommend publication of this work after the authors have addressed my following concerns:

(1) Some sentences are not very clear to me, for example, this sentence seems incomplete in the abstract: "...In the newly developed $(\text{K,Na})\text{NbO}_3$ solid-solution at the polymorphic phase boundary we combine atomic mapping of the local polar vector with in situ synchrotron X-ray diffraction and density functional theory..." What is the purpose of this combination of techniques? Furthermore, on page 4 of the manuscript, "...The above results underpin the coexistence of orthorhombic and tetragonal phases, which is observed not only at the level of average structure, but also in the atomic structure." What do you mean by "average" structure? And how can the coexistence of phases be observed in the atomic structure? Moreover, on page 13 of the manuscript, "...Experimental work was augmented by first-principles calculations revealing that the energy profiles of the two phases are very close and that the enthalpy is modified by the application of an electric field..." enthalpy will be modified by the application of an (electric) field with or without the augmentation of first-principle calculations, so the authors need to dig deeper into the findings by the first-principle calculations. Please proofread the manuscript carefully and make corrections.

(2) The authors need to rephrase the major findings of this work, which were written as "...The structural instability of ferroic materials with coexisting states of comparable energy is thus demonstrated pivotal in defining the high electromechanical properties in the quest for new

functional materials...” in the Abstract, and as “...Our study has verified that the field-induced phase transition can serve as an important structural instability to improve the electromechanical properties in ferroic materials...” in the Conclusion and Outlook. These are well-known statements in the field. The authors need to replace them with the unique scientific findings of this work and their impact to the field.

(3) Optional: The Authors may want to add some additional references to the “ultrahigh energy storage density” due to coexisting ferroic phases of comparable energies, for example, work by Cheng et al (“Demonstration of Ultra-high Recyclable Energy Densities in Domain-Engineered Ferroelectric Films” Nature Communications, 2017), and by Pan et al (“Ultrahigh energy storage in superparaelectric relaxor ferroelectrics”, Science, 2021).

Reviewer #3 (Remarks to the Author):

1) The authors prepared KNN based lead-free ceramics in terms of incorporating the BiFeO₃ with different contents. Their structures and the piezoelectric coefficients and strains were characterized. The local polar vectors were also atomically mapped using the in-situ synchrotron X-ray diffraction and density functional theory. In addition, the electric field induced phase transition approach was pursued to illustrate the large piezoelectric properties. A piezoelectric coefficient as large as 510 pm/V was measured, and an effective piezoelectric coefficient of 1250 pm/V was procured from the strain-electric field relationship along [200] direction.

2) The authors stated that the PPB existed at x=0.6 mol% of BiFeO₃ content, the point is if the BiFeO₃ acted as a main phase, or just as a dopant compound in which the Bi³⁺ and Fe³⁺ (Fe²⁺) ions entered into the lattice structure, the main crystalline phases (R and T) still came from the KNN?

3) If the Bi³⁺ and Fe³⁺ (Fe²⁺) ions entered into the lattice structure, what are the impacts of these ions on the piezoelectric coefficient?

4) It is suggested to show the P-E hysteresis loops for all the compositions in the supporting information.

5) It's better to present the loss tangents as a function of temperature and frequency for all the compositions in the supporting information.

6) To clarify the 1st or 2nd order phase transition, the permittivity as a function of temperature during the heating and cooling runs is suggested to be presented.

7) Some small errors, e.g., the PPB was defined again; while the STRAP did not defined; "Note also that" seems "note also that".

8) This paper can be accepted for publication with minor revision.

Response Letter

Thank you very much for handling our manuscript. Here, we would like to appreciate all the reviewers for their careful revision, constructive comments and the overall positive response. We have addressed all their comments below, carried out additional analysis and have prepared a revised version of the manuscript. The respective changes in the manuscript are highlighted in yellow.

Reviewer #1: The paper reports results of a synchrotron X-ray diffraction experiment and a density functional theory calculation of an electric field-induced phase transition of (K,Na)NbO₃-based lead-free piezoceramics with piezoelectric coefficients higher than 330 pC/N. The authors found that the phase transition between orthorhombic and tetragonal phases triggers a dramatic volume change and contributes to the high piezoelectricity at the polymorphic phase boundary. The results are informative and important to improve and develop high-performance lead-free piezoelectric materials. However, electric field-induced phase transitions at phase boundaries are well-known and have been extensively studied in other piezoelectric materials. For example, field-induced phase transitions in 0.70Pb(Mg_{1/3}Nb_{2/3})O₃-0.30PbTiO₃ piezoceramics have been demonstrated by in situ high-energy x-ray scattering (Phys. Rev. B 97, 214102 (2018)). In addition, the solid solution system reported in this study has a very complex chemical composition based on the known (K,Na)NbO₃ piezoelectric materials, and hence lack the novelty. I recommend that the paper should be published in another specialized journal such as Phys. Rev. B.

Reply: We thank the reviewer for carefully reading the manuscript and positively acknowledging the importance of our work.

Here, we would like to highlight the novelty and significance of our study from three different aspects:

- (1) **Fundamental difference in mechanism:** The mechanisms of the field-induced phase transitions of the studied materials are fundamentally different from the widely accepted theories describing the phase transition behavior of the Pb(Mg_{1/3}Nb_{2/3})O₃-PbTiO₃ (PMN-PT) system. First, the phase transition of PMN-PT proceeds via polarization rotation in monoclinic phases and is continuous in nature, as described in the paper mentioned by the reviewer (Hou *et al.* Phys. Rev. B, 97, 214102, 2018). However, the phase transition of the investigated materials (composition $x = 0.006$) is characterized by a discontinuous behavior near the coercive field, which is particularly evident in the 0°-sector where the electric field and the scattering vector are parallel. Due to such a field-induced phase transition, which occurs over a narrow excursion of

the electric field, and the accompanying structural instability, composition $x = 0.006$ can exhibit a large effective piezoelectric coefficient of 1250 pm/V along the $\langle 200 \rangle$ crystallographic direction. Second, polar nanoregions (PNRs) in PMN-PT ferroelectric relaxors play an important role (Li *et al.* Nat. Mater. 17, 349-354, 2018). PNRs account for 50%–80% of the dielectric and piezoelectric properties in PMN-PT materials (Li *et al.* Nat. Commun. 7, 13807, 2016 and Li *et al.* Adv. Fun. Mater. 27, 18, 1700310, 2017). However, PNRs as reported in PMN-PT crystals are not present in the studied KNN ferroelectrics of the current work.

- (2) **Nature of MPB:** The PMN-PT system mentioned by the reviewer is born with a morphotropic phase boundary (MPB) (Li *et al.* Nat. Commun. 7, 13807, 2016 and Park *et al.* J. Appl. Phys. 82, 1804, 1997). In contrast, such a phase boundary characterized by excellent electromechanical properties needs to be designed in lead-free KNN materials, which is one important achievement of the current work. To this end, we have developed the chemical compositions studied in this work by judicious modification of the composition and careful identification of the critical point in the vicinity of the phase boundary. As demonstrated in the manuscript, although two compositions, $x = 0.006$ and $x = 0.002$, are characterized by phase coexistence of tetragonal and orthorhombic phases, the field-induced phase transition and the associated structural changes are different for the two compositions and are particularly pronounced for composition $x = 0.006$. The neighboring composition, $x = 0.008$, does not feature a phase transition but non-180° domain switching, since the tetragonal phase dominates in this composition. Given the trivial differences (~ 0.1 mol% in BiFeO₃ content) between the three compositions, it is surprising that such a small difference can give rise to such a pronounced behavior of the field-induced phase transition. The above results highlight that the relevance of the phase boundary in lead-free KNN is far from being understood compared to the lead-containing compositions at the MPB.
- (3) **Contrast state-of-the-art of PZT to lead-free piezoceramics:** It has taken decades to develop the current understanding of the MPB in lead-based piezoelectrics. For example, lead-containing piezoelectrics such as Pb(Zr,Ti)O₃ already became widely used as early as in the 1960s. However, the theory of polarization rotation was first proposed in 1999 (Noheda *et al.* Appl. Phys. Lett., 74, 2059, 1999), which was followed by a series of studies that enriched the theoretical framework. This can be seen from the fact that each of the three papers by Noheda et al. in 1999/2000 have gained more

than 700 citations. Our study indicates that there is a lack of systematic understanding of the origins of the high functionalities in the phase boundary region in KNN-based materials and our present work constitutes an important step forward in filling this gap.

Reviewer #2: This is a good report which addressed the field-induced extrinsic piezoelectric responses in perovskite-type ferroelectrics. It used state-of-the-art characterization methods to clearly reveal the origin of the high piezoelectric response and the importance of the co-existence of phases of different symmetries with a composition near a phase boundaries. I recommend publication of this work after the authors have addressed my following concerns.

Reply: We thank the reviewer for the interest and the constructive comments, which will definitely improve this paper.

(1) Some sentences are not very clear to me, for example, this sentence seems incomplete in the abstract: "...In the newly developed (K,Na)NbO₃ solid-solution at the polymorphic phase boundary we combine atomic mapping of the local polar vector with in situ synchrotron X-ray diffraction and density functional theory..." What is the purpose of this combination of techniques?

Reply: Thanks for the question, the statement in the abstract has been revised as follows:

"In the newly developed (K,Na)NbO₃ solid-solution at the polymorphic phase boundary we combine atomic mapping of the local polar vector with in situ synchrotron X-ray diffraction and density functional theory to uncover the phase change and interpret its underlying nature".

Furthermore, on page 4 of the manuscript, "...The above results underpin the coexistence of orthorhombic and tetragonal phases, which is observed not only at the level of average structure, but also in the atomic structure." What do you mean by "average" structure? And how can the coexistence of phases be observed in the atomic structure?

Reply: Thanks for pointing this out. The phase coexistence of the investigated materials is experimentally confirmed by the structural characterizations at two different length scales, i.e. transmission electron microscopy (TEM) atomic mapping of the local polar vectors at the unit cell level and X-ray diffraction (XRD) experiments for the long-range ordered structure, with the correlated region large enough to produce sharp Bragg reflections. The latter is usually an average of a large amount of local-structural units and therefore is determined as the average structure (Keen and Goodwin, Nature, 521, 303, 2015). The corresponding sentence was modified as follows:

"The above results underpin the coexistence of orthorhombic and tetragonal phases, which is observed not only at the level of long-range ordered average structure, but is also confirmed by the atomic-scale structure and polar properties, indicating consistency between structures at different length scales".

Moreover, on page 13 of the manuscript, “...Experimental work was augmented by first-principles calculations revealing that the energy profiles of the two phases are very close and that the enthalpy is modified by the application of an electric field...” enthalpy will be modified by the application of an (electric) field with or without the augmentation of first-principle calculations, so the authors need to dig deeper into the findings by the first-principle calculations. Please proofread the manuscript carefully and make corrections.

Reply: We appreciate the reviewer’s insightful suggestion. Our first-principles calculations mark an important step forward in the development of mechanistic understanding of phase transitions driven by electric fields, which is expected to provide quantitative predictions to guide future experiments. Calculations are performed to evaluate the enthalpy (i.e., the Gibbs free energy at zero Kelvin) with the atomic positions optimized explicitly under external electric fields. These calculations are based on the formulation in the work of Souza *et al.* (Phys. Rev. Lett. 89, 117602, 2002) using the modern theory of electronic polarization (Resta *et al.* Rev. Mod. Phys. 66, 899, 1994). In particular, to our knowledge, there are only a few density functional theory codes (such as VASP, ABINIT, and CP2K) that can be used to evaluate the forces on atoms in insulating crystals with three-dimensional periodic boundary conditions while explicitly considering the external electric fields. Therefore, we are encouraged that the experimentally observed phase transitions driven by electric fields can be confirmed by our first-principles calculations. This motivates us to perform calculations on other materials and to extend the method for finite temperatures in the future, so that Gibbs free energies as a function of electric field and temperature can be explicitly evaluated at the density functional theory level. Correspondingly, following the reviewer’s suggestion, we went through the main text and modified the following sentence:

“Experimental work was supported by first-principles calculations demonstrating that the energy profiles of the two phases are very close and that the tetragonal phase is favored because of the lower enthalpy due to the application of a reverse electric field to the spontaneous polarization”.

In addition, additional experiments have been carried out so as to achieve a better combination of the first-principles calculations and experiments. For more details, please refer to our reply to the sixth comment from Reviewer 3.

(2) The authors need to rephrase the major findings of this work, which were written as “...The structural instability of ferroic materials with coexisting states of comparable energy is thus demonstrated pivotal in defining the high electromechanical properties in the quest for new functional materials...” in the Abstract, and as “...Our study has verified that the field-induced phase transition can serve as an important structural instability to improve the

electromechanical properties in ferroic materials...” in the Conclusion and Outlook. These are well-known statements in the field. The authors need to replace them with the unique scientific findings of this work and their impact to the field.

Reply: these are very detailed and relevant suggestions to improve the quality of the manuscript.

The statement in the abstract was modified as follows:

“The principle of using phase transition to promote electrostrain provides broader design flexibility in the development of high-performance piezoelectric materials and opens the door for the discovery of high-performance future functional oxides”.

The statement in the Conclusion and Outlook was changed as follows:

“Our study has demonstrated why the design of phase coexistence is of importance to obtain high-performance lead-free KNN piezoceramics and, more importantly, how structural instabilities at the phase boundary can be harvested to achieve an overall improved electromechanical response of the studied materials. Considering that not all boundary compositions guarantee the acquisition of optimal functional properties (e.g., composition $x = 0.002$ characterized by orthorhombic-tetragonal coexistence), our study also suggests that elaborate composition engineering is required for the identification of the critical composition in polycrystalline materials. This strategy of harnessing phase changes provides a path to a new paradigm for developing high-performance piezoceramics”.

(3) Optional: The Authors may want to add some additional references to the “ultrahigh energy storage density” due to coexisting ferroic phases of comparable energies, for example, work by Cheng et al (“Demonstration of Ultra-high Recyclable Energy Densities in Domain-Engineered Ferroelectric Films” Nature Communications, 2017), and by Pan et al (“Ultrahigh energy storage in superparaelectric relaxor ferroelectrics”, Science, 2021).

Reply: the two recommended references are very helpful and have been cited in the revised manuscript.

Reviewer #3:

(1) The authors prepared KNN based lead-free ceramics in terms of incorporating the BiFeO₃ with different contents. Their structures and the piezoelectric coefficients and strains were characterized. The local polar vectors were also atomically mapped using the in-situ synchrotron X-ray diffraction and density functional theory. In addition, the electric field induced phase transition approach was pursued to illustrate the large piezoelectric properties. A piezoelectric coefficient as large as 510 pm/V was measured, and an effective piezoelectric coefficient of 1250 pm/V was procured from the strain-electric field relationship along [200] direction.

Reply: We appreciate the reviewer’s careful reading of the manuscript.

(2) The authors stated that the PPB existed at $x=0.6$ mol% of BiFeO₃ content, the point is if the

BiFeO₃ acted as a main phase, or just as a dopant compound in which the Bi³⁺ and Fe³⁺+ (Fe²⁺) ions entered into the lattice structure, the main crystalline phases (R and T) still came from the KNN?

Reply: Thanks for the questions. The PPB in (K,Na)NbO₃ materials is defined as a temperature-driven phase boundary bridging the orthorhombic and tetragonal phases (Zhang *et al.* Appl. Phys. Lett., 91, 1 32913, 2007 and Wang *et al.* Appl. Phys. Lett., 95, 092905, 2009). The PPB is at 220 °C in pure KNN (Li *et al.* J. Am. Ceram. Soc. 96, 12, 3677, 2013). In this work, the incorporation of BiFeO₃ has two consequences. First, the addition of BiFeO₃ can shift the rhombohedral–tetragonal phase transition temperature from low temperature to room temperature (Zuo *et al.* J. Phys. Chem. Solids, 69, 230, 2008) because it crystallizes in rhombohedral symmetry with the space group *R3c* and thus stabilizes the rhombohedral phase in KNN. Second, the addition of BiFeO₃ shifts the orthorhombic to tetragonal phase transition temperature T_{O-T} from high temperature to room temperature without significantly reducing the Curie temperature (Zhou *et al.* J. Eur. Ceram. Soc. 32, 3575, 2012 and Wu *et al.* J. Mater. Chem. C, 8, 2838, 2020). As a result, a rhombohedral to tetragonal phase boundary can be built up near room temperature when a certain content of BiFeO₃ is reached (Wu *et al.* J. Am. Chem. Soc., 138, 15459, 2016), as depicted by the phase diagram in Figure R1.

Figure R1. Phase diagram of $(0.965-x)\text{K}_{0.5}\text{Na}_{0.5}\text{Nb}_{0.975}\text{Sb}_{0.25}\text{O}_3-0.035\text{Bi}_{1/2}\text{Na}_{1/2}\text{ZrO}_3-x\text{BiFeO}_3$ ceramics materials, obtained from Wu *et al.* J. Mater. Chem. C, 8, 2838, 2020.

In our work, compositions $x = 0.006$ and $x = 0.002$ are considered as the “PPB compositions” since they are characterized by a coexistence of the tetragonal phase and the orthorhombic phase at room temperature, as evidenced by the Rietveld refinement results (Supplementary Fig. 2). Since we have not found characteristic reflections of BiFeO₃ in our materials, we believe that BiFeO₃ acts as a dopant compound in the studied materials and both

Bi^{3+} and Fe^{3+} ions enter the lattice. Therefore, the KNN material is the host matrix and both the orthorhombic and tetragonal crystalline phases should originate from KNN.

Supplementary Fig. 2. Rietveld refinements of the high-energy X-ray diffraction pattern of composition $x = 0.006$. An enlarged view of the $\{110\}_{\text{PC}}$, $\{200\}_{\text{PC}}$, and high-angle reflections is also highlighted.

(3) If the Bi^{3+} and Fe^{3+} (Fe^{2+}) ions entered into the lattice structure, what are the impacts of these ions on the piezoelectric coefficient?

Reply: This is a very good question. We think that the two ions affect the piezoelectric properties mainly through their effects on the orthorhombic-tetragonal phase boundary of the materials. As described in the last answer, both compositions, $x = 0.006$ and $x = 0.002$, are close to the PPB. Nevertheless, composition $x = 0.006$ is the critical composition located at a more optimal position within the boundary region, which is due to the balanced phase fractions of the two phases. Based on the Rietveld refinement of the structures in the virgin state, the refined phase fraction of the tetragonal phase is 70% for composition $x = 0.006$ and 94% for composition $x = 0.002$. In particular, the phase fraction of the tetragonal phase of composition $x = 0.006$ varies between 47% and 70% during a complete cycle of the applied bipolar electric field. As a result, composition $x = 0.006$ exhibits better electromechanical properties and stronger structural instability upon the application of an electric field. Note that the dielectric maximum of composition $x = 0.006$ at the PPB is 2680, 25% higher than the value of 2150 for composition $x = 0.002$, as depicted in Figure R2. Since the dielectric constant (χ_{33}) and the piezoelectric coefficient (d_{33}) are intrinsically coupled ($d_{33} = 2\varepsilon_0\chi_{33}Q_{11}P_3$, Q_{11} is the electrostrictive coefficient and P_3 is the spontaneous polarization, Damjanovic, IEEE Trans. Ultrason. Ferroelectr. Freq. Control, 56, 8, 1574, 2009), the improved dielectric properties are

responsible for the better piezoelectric properties for composition $x = 0.006$.

Note that the dielectric anomaly for composition $x = 0.008$ is different from that for compositions $x = 0.002$ and $x = 0.006$ because such a content of BiFeO_3 merges the $T_{\text{R-O}}$ and $T_{\text{O-T}}$. The small dielectric anomaly observed at about -75°C for compositions $x = 0.002$ and $x = 0.006$, representing the transition from the rhombohedral to the orthorhombic phase, is not observed for composition $x = 0.008$.

Figure R2. Temperature dependence of the dielectric constant (left) and loss tangent ($\tan \delta$, right) of the investigated compositions in the poled state in the temperature range of -125 - 200°C , obtained at 1 kHz during the heating cycle.

(4) It is suggested to show the P-E hysteresis loops for all the compositions in the supporting information.

Reply: This is a very good suggestion. The polarization hysteresis loops and the unipolar strain loops of compositions $x = 0.002$, 0.006 , and 0.008 are now included in the supporting information as Supplementary Fig. 7.

Supplementary Fig. 7 (a) Polarization-electric field (P - E) hysteresis loops and (b) unipolar strain-electric field (S - E) hysteresis loops of compositions $x = 0.002$, 0.006 , and 0.008 , measured at a frequency of 0.1 Hz with a maximum field of 2 kV/mm in (a) and 1 kV/mm in (b).

(5) It's better to present the loss tangents as a function of temperature and frequency for all the compositions in the supporting information.

Reply: This is a good suggestion. The temperature dependence of the dielectric constant and loss tangent of the compositions $x = 0.002$, 0.006 , and 0.008 , quantified at 1 kHz– 1 MHz are included in the supporting information as Supplementary Fig. 8.

Supplementary Fig. 8 Temperature dependence of dielectric constant and loss tangent, $\tan \delta$, of compositions (a) $x = 0.002$, (b) 0.006 , and (c) 0.008 in the unpoled state and (d) $x = 0.002$, (e) 0.006 , and (f) 0.008 in the poled state, quantified at a frequency of 1 kHz– 1 MHz.

(6) To clarify the 1st or 2nd order phase transition, the permittivity as a function of temperature during the heating and cooling runs is suggested to be presented.

Reply: This is a very important suggestion, which led us to conduct a complementary latent heat study to clarify the nature of the phase transition. The temperature dependence of dielectric permittivity and loss tangent of composition $x = 0.006$ during the heating and cooling runs is displayed in Figure R3. The dielectric bump in the temperature range between 25 °C and 80 °C corresponds to the temperature-induced orthorhombic–tetragonal phase transition and the maximum at higher temperatures is the tetragonal–cubic phase transition. A very small thermal hysteresis is observed for both the dielectric permittivity and loss tangent in the heating and cooling runs. In contrast, the Curie point is characterized by an evident thermal hysteresis in the heating and cooling runs.

Figure R3. Temperature dependence of dielectric constant and loss tangent of composition $x = 0.006$ in the unpoled state in the heating and cooling runs, measured at a frequency of 10 kHz.

However, we think that the dielectric properties as function of temperature are not the most important clue for interpreting the nature of the field-induced orthorhombic–tetragonal phase transition observed in our materials. A first-order phase transition is characterized by the presence of latent heat, which is reflected in the change in sample temperature. And a second-order phase transition does not involve latent heat and temperature change. To this end, we have made an additional measurement to determine the temperature change of composition $x = 0.006$ when subjected to an applied electric field. An electric field of 1.5 kV/mm was slowly applied

to the sample, which was poled in opposite direction, and the electric field was held constant at maximum electric field to allow time for the system to relax back to ambient temperature, as depicted in the additional Figure 5b in the main text. The electric field was slowly increased at a rate of 0.012 kV/(mm·s) in order to completely diminish most of the electrocaloric-effect heat generated by the continuous polarization changes to the surroundings, thus allowing observations of the existing latent heat. The first anomaly with slight negative temperature change is related to the phase transition from a lower symmetry (orthorhombic) to a higher symmetry (tetragonal) where the latent heat is absorbed. The first anomaly maximizes at 0.37 kV/mm, which matches well to the coercive field (0.40 kV/mm) where the phase fraction of the tetragonal phase is maximized (Figure 2h in the manuscript). The second anomaly with positive temperature change is related to a combination of domain switching with heat generation due to friction of moving domain walls (Weyland *et al.* *J. Mater. Sci.*, 53, 9393, 2018) and the phase transition from a higher symmetry (tetragonal) to a lower symmetry (orthorhombic). However, note that a well-defined field-induced first-order phase transition is characterized by a very sharp change of sample temperature (Novak *et al.* *Phys. Rev. B*, 87, 104102, 2013 and Weyland *et al.* *Adv. Funct. Mater.* 26, 7326, 2016), which is different from the anomaly observed in this work. For example, a very sharp temperature change of 0.8 K was observed for BaTiO₃ single crystal, due to the field-induced cubic to tetragonal phase transition at 408 K, as featured in Figure R4. However, the anomaly becomes smeared and suppressed at 414 K, because the nature of the phase transition changes from first order to second order as it approaches the critical point (Kutnjak *et al.* *Nature*, 441, 04854, 2006, Novak *et al.* *Phys. Rev. B*, 87, 104102, 2013 and Wei *et al.* *Nat. Commun.* 12, 5322, 2021). The presence of the crossover from a first-order phase transition to continuous second-order phase transition is not unique in ferroelectric materials (Kutnjak *et al.* *Nature*, 441, 956, 2006) but a universal phenomenon in solid-state systems (Kerszberg *et al.* *Phys. Rev. Lett.*, 43, 293, 1979 and Folk *Phase Transit.*, 67, 645, 1999). In addition, it has been demonstrated that polycrystalline materials are characterized by a diffused critical point due to their inherent heterogeneity in grain orientation and are therefore not expected to have a sharply-defined critical point (Weyland *et al.* *Adv. Funct. Mater.* 26, 7326, 2016). Therefore, the second-order nature of the phase transition, which was predicted by the calculation without taking the application of an electric field into account, cannot be completely ruled out.

To briefly summarize, clarifying the nature of the phase transition (first order or second order) is relatively straightforward for single crystals (Weyland *et al.* *Adv. Funct. Mater.* 26, 7326, 2016), whereas it is very difficult for polycrystalline materials. Although this is not the

main focus of the current study, our combined DFT calculation and latent heat measurement suggest that the observed field-induced phase transition is likely to be second order.

Figure 5b. Quasi-isothermal calorimetric measurements at the field-induced phase transition of composition $x = 0.006$ at room temperature. A thermistor with a resolution within 0.1 mK was attached to the tested ceramic sample to detect the temperature change. The purple line describes the time dependence of the linearly changing bias electric field and the yellow line provides the change of sample temperature.

Figure R4. Change of sample temperature for BaTiO₃ single crystal due to released latent heat at the field-induced cubic to tetragonal phase transition, measured at several constant bath temperatures, obtained from Novak *et al.* Phys. Rev. B, 87, 104102, 2013.

The above discussion is now included in the revised manuscript:

“To clarify the nature of the phase transition, the temperature change of composition $x = 0.006$ is determined when an electric field is applied, since a first-order phase transition and a second-order phase transition can be distinguished by the presence of latent heat and the

associated change in sample temperature. An electric field of 1.5 kV/mm was slowly applied to the ceramic sample, which was poled in the opposite direction, and the electric field was held constant at maximum electric field to allow time for the system to relax back to ambient temperature, as depicted in **Figure 5b**. The first anomaly with a slight negative temperature change is related to the phase transition from the low-symmetry orthorhombic phase to the high-symmetry tetragonal phase where the latent heat is absorbed. It maximizes at 0.37 kV/mm, which matches perfectly to the coercive field where the phase transition is the most significant (**Figure 2h**). The second anomaly with positive temperature change is related to a combination of domain switching with heat generation due to friction of moving domain walls⁵³ and the transition from the tetragonal phase to the orthorhombic phase (**Figure 3b**). Note that a well-defined field-induced first-order phase transition is characterized by a very sharp change in sample temperature^{54, 55}. However, the observed temperature change is suppressed and smeared out, which is typically observed in the vicinity of the critical point, where a crossover from first-order to second-order phase transition takes places^{56, 57, 58}. It has been demonstrated that polycrystalline materials feature a diffuse critical point due to their inherent inhomogeneity in grain orientation, and are therefore not expected to have a sharply-defined critical point as in single crystals⁵⁴. Therefore, the second-order nature of the field-induced phase transition cannot be completely ruled out”.

(7) Some small errors, e.g., the PPB was defined again; while the STRAP did not defined; "Note also that" seems "note also that".

Reply: Thank you, these minor inadequacies have been corrected in the revised version. STRAP is the abbreviation of “Strain, Texture and Rietveld Analysis for Piezoceramics” (Hinterstein *et al.* Phys. Rev. B, 99, 174107, 2019) and the corresponding section has been updated as follows: “... calculated using a method that allows the analysis of all strain mechanisms with a Strain, Texture, and Rietveld Analysis for Piezoceramics (STRAP)⁴².”

(8) This paper can be accepted for publication with minor revision.

Reply: We are thankful for the positive feedback on the paper and the insightful comments, which have helped us improve the paper.

In addition to the above-mentioned changes requested by the reviewers, we have also made a series of minor modifications of the manuscript, in order to improve the language or the clarity of the presented results. All these are marked in yellow in the revised version.

REVIEWER COMMENTS

Reviewer #1 (Remarks to the Author):

The authors analyzed the XRD patterns using Rietveld refinements. However, only refined cell parameters and R_{w} are shown as fitting results in Supplementary Table 1. Other refined parameters (atomic coordinates, atomic displacement parameters, and fractions of the orthorhombic and tetragonal phases) and other reliable factors (R_I , R_{exp} , and goodness of fit) should be added.

The authors analyzed field-induced strain from the centers of mass of some Bragg reflections using equations (1-3). The field-induced lattice strain can be discussed in detail from the cell parameters of the coexisting orthorhombic and tetragonal phases determined by Rietveld refinements. I recommend that the authors add figures showing field dependences of the cell parameters of the coexisting orthorhombic and tetragonal phases.

The equation of the enthalpy shown in page 10 should be numbered. The enthalpy F and the unit cell volume V in the equation should be italic. The magnitude of the electric field is ϵ in the equation, whereas the electric field is E in figures 2-5. I recommend that ϵ is changed to E in the equation. In addition, the ground state energy E_0 in the equation is better to be changed to ϵ_0 to avoid misunderstandings.

Reviewer #2 (Remarks to the Author):

The authors have properly addressed all my concerns. I recommend publication of the revised manuscript in Nature Communications.

Reviewer #3 (Remarks to the Author):

1) For the reply to my comments, they are mainly acceptable. There is a small question on the field induced phase transition, and the heat generated. For the first-order phase transition, the latent heat should be added to the isothermal entropy change and adiabatic temperature change in terms of the Clausius-Clapeyron equation and the Maxwell relation. The authors may refer to the literature, J. Adv. Dielect. 2 (3) 1230011 (2012), and Crystals 10, 451 (2020) for details. For the field dependence of the adiabatic temperature change, a continuous transition was usually assumed, and ΔT is proportional to E^2 at low E-fields, and to $E^{2/3}$ at high E-fields. The authors may refer to the paper, Ceram. Inter. 41, S15 (2015) for details.

2) For the giant electrocaloric effect, it seems the paper should be mentioned besides Ref. 1), i.e., Science, 321, 821 (2008).

Response Letter

Thank you very much for inviting us to revise the manuscript for further consideration. Once a gain, we would like to sincerely thank all three reviewers for their careful reading, helpful suggestions, and overall positive feedback. We have addressed all of their comments below and have prepared a revised version of the manuscript. The respective changes in the manuscript are highlighted in yellow.

Reviewer #1:

(1) The authors analyzed the XRD patterns using Rietveld refinements. However, only refined cell parameters and R_w are shown as fitting results in Supplementary Table 1. Other refined parameters (atomic coordinates, atomic displacement parameters, and fractions of the orthorhombic and tetragonal phases) and other reliable factors (R_I , R_{exp} , and goodness of fit) should be added.

Reply: This is a very good suggestion. The refined parameters, including atomic coordinates, atomic displacement parameters, and phase fractions of all three compositions, are now included in the revised supplementary file:

Supplementary Table 1. Refined cell parameters of the compositions $x = 0.002$, $x = 0.006$, and $x = 0.008$.

	$x = 0.002$	$x = 0.006$	$x = 0.008$
R_{wp} (%)	7.83	10.03	7.86
R_{exp} (%)	4.03	2.83	4.25
GOF (χ^2)	1.94	3.54	1.85
Amm2			
Weigh fraction	0.8847(5)	0.5367(3)	-
a (Å)	3.96978(23)	3.96472(25)	-
b (Å)	5.6439(16)	5.6328(7)	-
c (Å)	5.6376(20)	5.6456(5)	-
A-site x	0	0	-
A-site y	0	0	-
A-site z	0	0	-
B-site x	0.5	0.5	-
B-site y	0	0	-
B-site z	0.4649(6)	0.4743(1)	-
O(1) x	0	0	-
O(1) y	0	0	-
O(1) z	0.4274(0)	0.4352(2)	-

O(2) x	0.5	0.5	-
O(2) y	0.7535(0)	0.8027(8)	-
O(2) z	0.1832(0)	0.1963(8)	-
A-site U _{iso} (Å ²)	0.0109(2)	0.0139(7)	-
B-site U _{iso} (Å ²)	0.0075(8)	0.0069(1)	-
O-site U _{iso} (Å ²)	0.0066(5)	0.0016(2)	-
P4mm			
Weight fraction	0.1152(5)	0.4632(7)	1
a (Å)	3.97277(4)	3.97195(16)	3.94450(33)
c (Å)	4.00161(7)	3.99594(20)	4.00342(6)
A-site x	0	0	0
A-site y	0	0	0
A-site z	0	0	0
B-site x	0.5	0.5	0.5
B-site y	0.5	0.5	0.5
B-site z	0.4821(0)	0.4757(2)	0.4823(1)
O(1) x	0.5	0.5	0.5
O(1) y	0.5	0.5	0.5
O(1) z	-0.0394(0)	0.0048(1)	-0.0494(3)
O(2) x	0.5	0.5	0.5
O(2) y	0	0	0
O(2) z	0.4532(9)	0.4799(7)	0.4566(9)
A-site U _{iso} (Å ²)	0.0187(5)	0.0322(6)	0.0253(9)
B-site U _{iso} (Å ²)	0.0056(5)	0.0079(0)	0.0063(5)
O-site U _{iso} (Å ²)	0.0187(7)	0.0185(0)	0.0177(10)

Reliable factors, including the weighted profile *R*-factor, R_{wp} , the expected *R*-factor R_{exp} , and goodness of fit (χ^2) are provided. The above factors are the most important and valuable factors as measures of refinement factor (Toby *et al.* Powder Diffr., 21, 67, 2006). The intensity-based *R* factor, R_I , defined as $R_I = (\sum_{hkl} I_{O,hkl} - I_{C,hkl}) / (\sum_{hkl} I_{O,hkl})$ (Toby *et al.* Powder Diffr., 21, 67, 2006), is not commonly used and is therefore not provided here.

(2) The authors analyzed field-induced strain from the centers of mass of some Bragg reflections using equations (1-3). The field-induced lattice stain can be discussed in detail from the cell parameters of the coexisting orthorhombic and tetragonal phases determined by Rietveld refinements. I recommend that the authors add figures showing field dependences of the cell parameters of the coexisting orthorhombic and tetragonal phases.

Reply: Thanks for the suggestion. The field dependence of the cell parameters of the orthorhombic and tetragonal phases for the compositions $x = 0.006$ and $x = 0.002$, both characterized by phase coexistence, is shown in Supplementary Fig. 5 and

Supplementary Fig. 6, respectively. The evolution of cell parameters with field of the composition $x = 0.006$ is more significant than that of the composition $x = 0.002$, indicating that the former is structurally more active. For example, the cell parameter b_0 decreases while the cell parameter c_0 remains almost unchanged with increasing electric field. This indicates that the in-plane orthorhombic lattice distortion (c_0/b_0) increases with increasing electric field. Meanwhile, c_T decreases, while a_T increases with increasing electric field, indicating that the tetragonal lattice distortion (c_T/a_T) decreases with increasing electric field. The above results suggest that the orthorhombic lattice distortion is favored while the tetragonal lattice distortion is suppressed with increasing electric field. This is consistent with the observation that the orthorhombic phase is favored at high electric field, as evidenced by the phase fraction of the two phases with increasing electric field (**Figure 2** in the manuscript).

Although we completely agree with the reviewer that it might be a good idea to analyze the field-induced lattice strain from the perspective of the cell parameters of two coexisting phases, we would like to emphasize that the evolution of cell parameters with field is a measure of the average structure averaged over the entire sample, whereas the field-induced lattice strain has a pronounced anisotropic character, as shown in **Figure 3** in the manuscript. Therefore, in our case it is more practical to interpret the field-induced strain using the center-of-mass method, since it allows us to quantify the effective piezoelectric response for grains with different orientations. In particular, the most significant structural change is recorded along the $\langle 200 \rangle$ directions for the composition $x = 0.006$, which is due to the field-induced phase transition that occurs over a narrow excursion of the field parallel to $\langle 200 \rangle$.

At the suggestion of the reviewer, the field dependence of cell parameters has now been included in the revised manuscript.

Supplementary Fig. 5. Evolution of lattice parameters of coexisting **a** orthorhombic and **b** tetragonal phases for the composition $x = 0.006$ as a function of bipolar electric field,

calculated by the method STRAP (Strain, Texture, and Rietveld Analysis for Piezoceramics) (Hinterstein *et al.* Phys. Rev. B, 99, 174107, 2019).

Supplementary Fig. 6. Evolution of the lattice parameters of the **a** orthorhombic and **b** tetragonal phases for the composition $x = 0.006$ as a function of bipolar electric field, calculated using the STRAP (Strain, Texture, and Rietveld Analysis for Piezoceramics) method (Hinterstein *et al.* Phys. Rev. B, 99, 174107, 2019).

(3) The equation of the enthalpy shown in page 10 should be numbered. The enthalpy F and the unit cell volume V in the equation should be italic. The magnitude of the electric field is ε in the equation, whereas the electric field is E in figures 2-5. I recommend that ε is changed to E in the equation. In addition, the ground state energy E_0 in the equation is better to be changed to ε_0 to avoid misunderstandings.

Reply: we appreciate the reviewer’s careful reading of the manuscript and the corresponding section has been updated as follows:

$$F = \varepsilon_0 - VP \cdot E \quad (4)$$

where ε_0 is the ground state energy, V is the unit cell volume, P is the macroscopic polarization, and E is the magnitude of the electric fields.”

Reviewer #2: The authors have properly addressed all my concerns. I recommend publication of the revised manuscript in Nature Communications.

Reply: We thank the reviewer for the interest and support for this work. The constructive comments and helpful suggestions have helped us to improve this manuscript.

Reviewer #3:

(1) For the reply to my comments, they are mainly acceptable. There is a small question on the field induced phase transition, and the heat generated. For the first-order phase transition, the latent heat should be added to the isothermal entropy change and adiabatic temperature change in terms of the Clausius-Clapeyron equation and the Maxwell relation. The authors may refer to the literature, J. Adv. Dielect. 2 (3) 1230011 (2012), and Crystals 10, 451 (2020) for details.

For the field dependence of the adiabatic temperature change, a continuous transition was usually assumed, and ΔT is proportional to E^2 at low E-fields, and to $E^{2/3}$ at high E-fields. The authors may refer to the paper, *Ceram. Inter.* 41, S15 (2015) for details.

Reply: We thank the reviewer for careful reading of the manuscript and important suggestions. It should be noted that our thermometric measurement was performed under the condition that the studied sample was subjected to an electric field at a very slow rate of 0.012 kV/(mm·s) to avoid adiabatic conditions and thus completely reduce the influence of the electrocaloric effect. In the literature, to directly measure the adiabatic temperature change of the electrocaloric effect, an electric field is sharply applied or removed with a rate of 100 kV/(mm·s) or even higher, as described in detail in the references recommended by the reviewer (e.g., Lu *et al.*, *Crystals*, 10, 451, 2020). According to previous studies on similar phase transitions induced by an electric field (Novak *et al.* *Phys. Rev. Lett.*, 109, 037601, 2012 and Novak *et al.* *Phys. Rev. B*, 97, 094113, 2018), the measured temperature change can be primarily attributed to the release of latent heat due to the phase transition upon application of a slow electric field. The Clausius-Clapeyron equation and Maxwell relation are most commonly used to determine the electrocaloric effect rather than just the latent heat (Moya *et al.* *Nat. Mater.*, 13, 439, 2014). The recommended references are relevant and have been cited in the revised manuscript. The corresponding section in the manuscript is revised as follows:

“An electric field of 1.5 kV/mm was slowly applied to the ceramic sample to avoid the temperature change caused by the electrocaloric effect^{54, 55, 56}”.

(2) For the giant electrocaloric effect, it seems the paper should be mentioned besides Ref. 1), i.e., *Science*, 321, 821 (2008).

Reply: The recommended reference is an important work, which has been cited in the revised manuscript.

REVIEWERS' COMMENTS

Reviewer #1 (Remarks to the Author):

The caption of the revised Supplementary Table 1 should be also changed. The added GOF of the Rietveld analysis for $x = 0.006$ sample is higher than those for other samples. O-site Uiso = 0.0066 and 0.0016 of the orthorhombic phases are unusually smaller than O-site Uiso = 0.0187 and 0.0185 of the tetragonal phases in $x = 0.002$ and 0.006 samples. A-site Uiso = 0.0322 of the tetragonal phase is unusually larger than A-site Uiso = 0.0139 of the orthorhombic phase in $x = 0.006$ sample. Please address these issues, and carefully check and/or reanalyze all of the data.

Reviewer #3 (Remarks to the Author):

The authors have answered all my concerns. The manuscript can be accepted for publication in Nature Communications now.

Response Letter

Thank you very much for giving us the opportunity to address the remaining concerns of the reviewers before the manuscript is accepted for publication in your journal. We believe that the quality of the manuscript has been improved after three rounds of peer review, which would not be impossible without the careful reading and insightful suggestions of three professional reviewers. We have addressed the comments as follows and have prepared a revised version of the manuscript. The respective changes in the manuscript are highlighted in yellow.

Reviewer #1:

The caption of the revised Supplementary Table 1 should be also changed. The added GOF of the Rietveld analysis for $x = 0.006$ sample is higher than those for other samples. O-site $U_{iso} = 0.0066$ and 0.0016 of the orthorhombic phases are unusually smaller than O-site $U_{iso} = 0.0187$ and 0.0185 of the tetragonal phases in $x = 0.002$ and 0.006 samples. A-site $U_{iso} = 0.0322$ of the tetragonal phase is unusually larger than A-site $U_{iso} = 0.0139$ of the orthorhombic phase in $x = 0.006$ sample. Please address these issues, and carefully check and/or reanalyze all of the data.

Reply: We really appreciate the reviewer's careful reading of the manuscript and very helpful suggestions. We have reanalyzed the structural data of the compositions $x = 0.002$ and $x = 0.006$, as indicated in the revised Supplementary Table 1. In particular, the thermal parameters, U_{iso} , of the A-site, B-site, and O-site cations of the two coexisting phases are comparable. The slightly higher GOF for the composition $x = 0.006$ is due to the slightly higher R_{wp} value, since $GOF^2 = \chi^2 = (R_{wp}/R_{exp})^2$ (Toby *et al.* Powder Diffr., 21, 67, 2006). However, the quality of the Rietveld fit for this composition is as good as for the other two compositions, as evidenced by the overall good agreement between the observed and calculated patterns. We also tried to improve the model and managed to lower the R_{wp} value from 10.03% to 9.93%.

Supplementary Table 1. Results of Rietveld refinement for the compositions $x = 0.002$, $x = 0.006$, and $x = 0.008$. Refined structural parameters of compositions $x = 0.002$ and $x = 0.006$ with *Amm2* and *P4mm* models and refined structural parameters of composition $x = 0.008$ with *P4mm* model.

	$x = 0.002$	$x = 0.006$	$x = 0.008$
R_{wp}	7.32	9.93	7.86
R_{exp}	4.04	2.86	4.25
GOF (χ^2)	1.81	3.47	1.85
Amm2			
Percentage (%)	81.76(7)	67.28(9)	-

a (Å)	3.96995(27)	3.96590(13)	-
b (Å)	5.6424(11)	5.6328(4)	-
c (Å)	5.6413(13)	5.64550(27)	-
A-site x	0	0	-
A-site y	0	0	-
A-site z	0.0194(1)	0.0419(7)	-
B-site x	0.5	0.5	-
B-site y	0	0	-
B-site z	0.4853(3)	0.5165(3)	-
O(1) x	0	0	-
O(1) y	0	0	-
O(1) z	0.5560(6)	0.4592(1)	-
O(2) x	0.5	0.5	-
O(2) y	0.7925(3)	0.7939(7)	-
O(2) z	0.1759(0)	0.2402(8)	-
A-site U_{iso}	0.0152(3)	0.0273(3)	-
B-site U_{iso}	0.0071(5)	0.0068(9)	-
O-site U_{iso}	0.0136(1)	0.023(1)	-

P4mm

Percentage (%)	18.23(3)	32.71(1)	100
a (Å)	3.97264(6)	3.97166(15)	3.94450(33)
c (Å)	4.00191(10)	3.99766(19)	4.00342(6)
A-site x	0	0	0
A-site y	0	0	0
A-site z	0	0	0
B-site x	0.5	0.5	0.5
B-site y	0.5	0.5	0.5
B-site z	0.4850(2)	0.4674(30)	0.4823(1)
O(1) x	0.5	0.5	0.5
O(1) y	0.5	0.5	0.5
O(1) z	-0.0260(7)	-0.0002(1)	-0.0494(3)
O(2) x	0.5	0.5	0.5
O(2) y	0	0	0
O(2) z	0.4565(4)	0.4990(3)	0.4566(9)
A-site U_{iso}	0.0171(1)	0.0261(8)	0.0253(9)
B-site U_{iso}	0.0055(7)	0.0071(1)	0.0063(5)
O-site U_{iso}	0.010(7)	0.021(6)	0.0177(10)

Reviewer #3:

The authors have answered all my concerns. The manuscript can be accepted for publication in Nature Communications now.

Reply: We thank the reviewer for the interest and support for this work. The constructive comments and helpful suggestions have helped us to improve this manuscript.